# Frozen Accident Pushing 50: Stereochemistry, Expansion, and Chance in the Evolution of the Genetic Code

**DOI:** 10.3390/life7020022

**Published:** 2017-05-23

**Authors:** Eugene V. Koonin

**Affiliations:** National Center for Biotechnology Information, National Library of Medicine, National Institutes of Health, Bethesda, MD 20894, USA; koonin@ncbi.nlm.nih.gov; Tel.: +1-301-435-5913

**Keywords:** genetic code evolution, frozen accident, stereochemical theory, coevolution theory, error minimization, RNA world, proto-tRNAs

## Abstract

Nearly 50 years ago, Francis Crick propounded the frozen accident scenario for the evolution of the genetic code along with the hypothesis that the early translation system consisted primarily of RNA. Under the frozen accident perspective, the code is universal among modern life forms because any change in codon assignment would be highly deleterious. The frozen accident can be considered the default theory of code evolution because it does not imply any specific interactions between amino acids and the cognate codons or anticodons, or any particular properties of the code. The subsequent 49 years of code studies have elucidated notable features of the standard code, such as high robustness to errors, but failed to develop a compelling explanation for codon assignments. In particular, stereochemical affinity between amino acids and the cognate codons or anticodons does not seem to account for the origin and evolution of the code. Here, I expand Crick’s hypothesis on RNA-only translation system by presenting evidence that this early translation already attained high fidelity that allowed protein evolution. I outline an experimentally testable scenario for the evolution of the code that combines a distinct version of the stereochemical hypothesis, in which amino acids are recognized via unique sites in the tertiary structure of proto-tRNAs, rather than by anticodons, expansion of the code via proto-tRNA duplication, and the frozen accident.

## 1. Introduction

The time of this writing, early 2017, falls between two notable dates, the 100th anniversary of Francis Crick’s birth and the 50th anniversary of Crick’s 1968 classic paper on the evolution of the genetic code [1,2]. Compared to Crick’s momentous contribution to the understanding of DNA structure and replication [3,4], and then the principles of protein translation [5,6], the code evolution paper might seem to be almost inconsequential. Yet, this masterpiece of conceptual thinking presents two ideas that have shaped the subsequent developments in the study of the code evolution and more generally, the study of the early evolution of life. These ideas are the frozen accident perspective on the code evolution and the inference of a RNA-only translation system. 

The genetic code that defines the rules of translation from the 4-letter nucleic acid alphabet to the 20-letter alphabet of proteins is arguably the single central informational invariant of all life forms [6,7,8,9]. Indeed, notwithstanding multiple code variants that continue to emerge through the study of protein coding in diverse life forms, the basic structure of the code and the majority of the codon assignments are truly universal [10,11]. When the codon table was settled in 1965, distinct patterns in the code begging for explanation became immediately apparent [9,12]. The 64 triplet codons are neatly organized in sets of four or two, with the third base of a codon typically being synonymous. The assignment of codons to amino acids across the code table is clearly non-random: related amino acids typically occupy contiguous areas in the table. The second position of a codon is the most important specificity determinant, and three of the four columns of the table encode related, chemically similar amino acids. For example, all codons with a U in the second position correspond to hydrophobic amino acids. It is obvious from the table itself that the code is robust to error, i.e., mutational and translation errors in synonymous positions (typically, the third position in a codon) have no effect on the protein, whereas substitutions in the first position typically lead to incorporation of an amino acid similar to the correct one, thus decreasing the damage [9]. Quantitative analyses of the code using cost functions derived from physico-chemical properties of amino acids or their evolutionary exchangeability have confirmed the exceptional robustness of the standard genetic code (SGC): the probability to reach the same level of error minimization as in the SGC by random permutation of codons is below 10^−6^ [10,13,14,15]. However, the SGC is far from being optimal because, given the enormous overall number of possible codes (>10^84^), billions of variants are even more robust to error [10]. 

During the 49 years since the publication of Crick’s code evolution paper, extensive experimental and theoretical research effort had aimed to develop a definitive scenario for the evolution of the code that would account for its notable properties. Even if rarely coached that way, these studies can be considered as attempts on falsification of the frozen accident scenario. Although the features of the code have been explored in detail and some aspects of its evolution seem to have been elucidated, these efforts appear to fall short of a compelling refutation of the frozen accident. 

In this concept article, I briefly review the current understanding of the factors that determined the universality of the SGC and the three main scenarios of the code evolution. I then sketch a new scenario for code evolution, informed by comparative genomic analyses, that combines the idea of a primordial stereochemcial code with the frozen accident perspective. As a disclaimer, I should note that no attempt is made here on anything close to a comprehensive review of the research on the code origin and evolution let alone the origin of life. The goal is to place the frozen accident concept into the context of latest efforts in the field and briefly discuss some new ideas.

## 2. Why the Universal Code?

The genetic code is nearly universal in all extant life forms although limited deviations from the SGC have been detected in many groups of organisms, particularly in organelles and parasitic or endosymbiotic bacteria with highly reduced genomes [11,16,17]. The changes to the SGC follow three distinct patterns: (i) reassignment of codons within the canonical set of 21 (including the stop signal); (ii) loss (“unassignment”) of codons, and (iii) incorporation of new amino acids. Stop codons are strongly over-represented among the code modifications. Of the 23 non-standard code variants listed in a recent survey [11], there are 8 cases of stop codons being reassigned or acquired, 8 cases of codon loss, and 10 reassignments of a codon from one amino acid to another. Given that there are only three stop codons, the same changes to the code occurred in parallel in different groups of organisms. 

At least two well-characterized non-canonical amino acids have been co-opted into the code, namely, selenocysteine that is represented in varying sets of proteins in diverse organisms from all three domains of life [18,19], and pyrrolysine, currently detected only in some archaea. The mechanisms that lead to the incorporation of these two amino acids are completely different. Pyrrolysine is accommodated via reassignment of a stop codon analogous to code changes within the canonical amino acid set; in contrast, inclusion of selenocysteine involves recoding whereby a stop codon directs selenocysteine incorporation only in the presence of a distinct, regulatory sequence element [20,21,22].

The evolutionary mechanisms that lead to codon reassignment and emergence of deviant codes involve changes in tRNA specificity and/or evolution of new specificities in the case of stop codon recruitment. These mechanisms fit the general ‘gain and loss’ framework, where ‘gain’ refers to acquisition of a new tRNA specificity, typically resulting from duplication of a tRNA gene, whereas ‘loss’ is elimination of a tRNA specificity, typically via deletion [11,23,24]. There is no evidence that any modern code modifications are adaptive. Most likely, these change evolved neutrally, through genetic drift and mutational pressure that drives small genomes toward high AT-content. The single key feature to be emphasized regarding the extant code variants is that they are all minor, i.e., involve one or at most two reassignments per variant, and typically concern rare amino acids, such as reassigning tryptophane (the least abundant amino acid of all) to a stop codon, the most common change that occurred in parallel in several deviant codes. The variant code does not venture far from the SGC at all.

Notably, new methods of synthetic biology have recently allowed substantial artificial alteration of the code in bacteria [25,26,27,28]. Although the fitness of the bacteria with altered codes has not been thoroughly studied, their viability itself seems to support the view that the fitness of different codes might not differ dramatically, so that the (near) universality of the SGC is likely to stem from high fitness barriers separating it from other codes, or in other words, low fitness of the intermediates [11].

To account for the universality of the code, Crick came up with the frozen accident argument: “This theory states that the code is universal because at the present time any change would be lethal, or at least very strongly selected against. This is because in all organisms (with the possible exception of certain viruses) the code determines (by reading the mRNA) the amino acid sequences of so many highly evolved protein molecules that any change to these would be highly disadvantageous unless accompanied by many simultaneous mutations to correct the “mistakes” produced by altering the code. This accounts for the fact that the code does not change. To account for it being the same in all organisms one must assume that all life evolved from a single organism (more strictly, from a single closely interbreeding population). In its extreme form, the theory implies that the allocation of codons to amino acids at this point was entirely a matter of “chance”.” [1].

Using the language of fitness landscapes, the frozen accident perspective implies that there can be many different codes occupying fitness peaks, but they are separated by deep valleys of low fitness [15] (Figure 1). As indicated above, all discovered codon reassignments in extant organisms are indeed quite limited in scope. Moreover, in agreement with Crick’s reasoning, these modifications have been identified primarily in organelles and bacteria with small genomes where the damage from reassigning rare codons could have been tolerable (contrary to what Crick thought, the code apparently does not change in viruses because these employ the host translation machinery for decoding their mRNAs). The frozen accident argument does not necessarily require that the original choice of codon assignment is literally and strictly random. Various factors could have contributed to the initial codon assignments (see discussion below) but once the choice is made, it gets frozen, i.e., only rare and minor changes may be allowed.

As Crick points out, the universality of the code is perhaps the strongest evidence of the existence of LUCA (Last Universal Cellular Ancestor), along with the universal conservation of the translation machinery [29]. However, LUCA certainly was not the first life form and most likely, not even the first cellular organism, only an evolutionary bottleneck. The early stages of cellular and especially pre-cellular evolution must have been dramatically different from the post-LUCA evolution of cellular organisms. This early evolution is thought to have involved competition between ensembles of “virus-like” genetic elements and selection at the level of such collectives [30,31,32]. Translation hardly could have evolved literally within a single such ensemble, and if it emerged on multiple occasions in different ensembles, then, initially, numerous, different codes must have existed. Then, why only one frozen accident, assuming that the actual codon assignments are indeed (quasi) accidental? To put it even more generally, why a single LUCA only?

A strikingly simple but, I believe, compelling answer was given by Vetsigian, Woese, and Goldenfeld in a breakthrough 2006 paper [33]: only one code survived because extensive horizontal gene transfer (HGT) was an aspect of early evolution without which the transition to the cellular level of complexity would not have been possible. Even a small change in the code has a prohibitive effect on HGT. Vetsigian and colleagues developed a simple mathematical model to track evolution with and without HGT. Starting with a random ensemble of codes, their simulated evolution experiments led to increased coded diversity in the absence of HGT but to survival of only a few code variants when HGT was allowed. In the original study, Vetsigian and coworkers explored a deterministic model in an infinite population approximation [33]. Notably, recent modeling efforts that took into account stochastic effects of the finite population size produced a single, universal code, with a structure similar to that of the SGC, within a range of HGT rates [34,35].

Given that HGT, even if limited by both physical and selective barriers, remained a key factor of microbial evolution [36,37,38,39,40], and apparently, a condition for long term survival of microbial populations [41,42], it stands to reason that it played a key role not only in the primordial universalization of the code but also in its maintenance through nearly four billion years of cellular evolution. The demonstration of the essential role of the code universality in the exchange of genetic information that promotes evolution of life arguably was one of the most important insights in the code evolution field since Crick’s hypothesis. Conceivably, frozen accident and the requirement of HGT were major, complementary factors that have kept the code universal.

## 3. The Three Principal Scenarios of the Code Origin and Evolution: Achievements, Limitations, and Compatibility

The structure of the SGC, i.e., the mapping of 64 codons to 20 amino acids and the stop signal, is clearly non-random by multiple criteria [9,10]. This non-randomness of the code seems to require an explanation(s). The three major concepts that strive to explain the regularities in the code are the stereochemical, coevolution, and error minimization ‘theories’ [10,43,44]. A detailed review of the current state of these scenarios for code evolution is presented elsewhere [45]; here, I only give a brief synopsis and assessment of their contributions to our current understanding of the code evolution. 

The stereochemical theory postulates that the structure of the code is determined by physicoo-chemical affinity between amino acids and codons or anticodons. The initial attempts on direct experimental demonstration of interaction between amino acids and the cognate triplets have been generally unconvincing [46]. In these early days of the study of code evolution, molecular modeling has been used to propose a variety of ‘stereochemical codes’ based on interactions of amino acids with cognate codons [47], reversed codons [48], anticodons [49], codon-anticodon duplexes [50], or a complex of four nucleotides including the anticodon and an additional base [51]. The difficulty of discriminating between different models and the lack of solid experimental support prevented any of these schemes from becoming the explanatory framework for the evolution of the code.

The stereochemical theory has been given a new life by the progress of the aptamer technology. At least some amino acids have been shown to select sequences significantly enriched for either codons or anticodons [52,53,54,55]. These results were taken as evidence of existence of an early stereochemical era in the code evolution during which the majority of the modern amino acids have been co-opted into the code [56]. However, as discussed elsewhere [10,45], the stereochemical evidence does not appear to be compelling. The principal problems are two-fold. First, statistically significant affinity for cognate triplets has been demonstrated only for five amino acids, and the triplet could be either codon or anticodon [55]. Second, and more damning, all the results of aptamer experiments that appear compatible with the physico-chemical affinity between amino acids and cognate triplets pertain to complex amino acids that are unlikely to have been available at the time of the code emergence (see next Section).

The second theory, known as coevolution, holds that the genetic code is shaped by the precursor-product relationships between amino acids [57,58,59,60]. Under the coevolution scenario, the code evolved from an ancestral version that included only the simple amino acids that can be produced abiogenically (see next section) by expanding to incorporate the more complex amino acids, in parallel with the evolution of the respective biosynthetic pathways (hence coevolution—between the code and the amino acid biosynthesis pathways). The importance of biosynthetic pathways for the code evolution is effectively obvious because amino acids could not be incorporated into the code unless they were available. The coevolution theory holds that the code evolved by subdivision of large blocks of codons that in the ancestral code encoded the same amino acid but were split to encode two (or more) amino acids upon the evolution of the respective metabolic pathways. The specific pattern of codon reassignment would be determined by the precursor-product relationships between amino acids. Thus, the coevolution theory cannot be reduced to the self-evident statement on the importance of biosynthetic pathways for the inclusion of the late amino acids into the code. Rather, the theory makes specific and readily falsifiable predictions on the subdivision of the primordial blocks of codons assignments following the evolution of metabolic pathways. As shown within the framework of the third theory of code evolution, these inferences do not seem to hold very well. 

The grouping of similar amino acids within the same column of the code table immediately indicates that the code has a degree of robustness to mutational and translational errors. In other words, the codon assignments are organized in such a way as to minimize the deleterious effect of such errors. Hence the error minimization theory of code evolution, under which the structure of the code is determined by selection for robustness to errors. Extensive quantitative analyses that employed cost functions differently derived from physico-chemical properties of amino acids have shown that the code is indeed highly resilient, with the probability to pick an equally robust random code being on the order of 10^−7^–10^−8^ [14,15,61,62,63,64,65,66,67,68]. Obviously, however, among the ~10^84^ possible random codes, there is a huge number with a higher degree of error minimization than the SGC. Furthermore, the SGC is not a local peak on the code fitness landscape because certain local rearrangements can increase the level of error minimization; a quantitatively, the SGC is positioned roughly halfway from an average random code to the summit of the corresponding local peak [15] (Figure 1).

The typical conclusion of the error minimization theorists is that the code evolved under selective pressure for robustness. However, this conclusion is not necessarily justified. Most if not all code optimization analyses focus on rearrangements of the standard code table, which allows formal (and in itself, valid, given the cost function) estimation of robustness but not the evolutionary processes that lead to it. A more biologically sound approach involves reconstruction of the routes of code expansion that might produce error minimization as a selectively neutral byproduct of evolution driven by other factors [64,67,68].

To conclude this brief discussion of the three major scenarios for the evolution of the code, it is useful to note that, although stemming from widely different premises, there is no reason why these scenarios should be mutually exclusive. Quite the contrary, stereochemistry, biochemical coevolution, and selection for error minimization could have contributed synergistically at different stages of the evolution of the code [43]—along with frozen accident. 

## 4. Primordial Expansion of the Code

Little as we can claim to actually know about the evolution of the code, it appears most likely that the earliest proteins contained fewer amino acids than the modern set of 20. The canonical amino acids differ substantially in terms chemical complexity and stability. Ten amino acids have been consistently identified in prebiotic chemistry experiments as well as in in meteorites, in the following order of abundance: Gly, Ala, Asp, Glu, Val, Ser, Ile, Leu, Pro, Thr [69,70,71,72]. Notably, the recent, perhaps most promising at this time prebiotic chemistry experiments, based on hydrogen cyanide photochemistry, that yield precursors of ribonucleotides and amino acids, also primarily produce amino acids from the above list, namely, Gly, Ala, Ser, and Thr [73,74]. The ranks of amino acids in the ‘early’ list positively and significantly correlate with the free energy of their synthesis: the amino acids that are on top of the list are the ‘cheapest’ energetically [75]. An independent approach that involved analysis of the fluxes of amino acids in recent evolution of proteins in diverse organisms shows that the concentrations of the putative early amino acids in the above list are mostly decreasing, whereas those of the late amino acids are increasing [76]. This convergence of widely different methods of inference suggests that the above 10 amino acids can be confidently considered old, i.e., were represented already in the first proteins [77].

Given the confidently derived consensus set of ancient amino acids and the unavailability of the late amino acids before complex biosynthetic pathways evolved (see discussion of the coevolution theory above), early expansion of the code appears to be inevitable. The idea of such expansion again goes back to Crick’s seminal paper: “The next general point about the primitive code is that it seems likely that only a few amino acids were involved” [1]. More importantly, Crick proposed, if not quite explicitly, that the error minimization property of the code could have evolved as consequence of the code expansion: “the net effect of a whole series of such changes would be that similar amino acids would tend to have similar codons, which is just what we observe in the present code” [1]. 

The theme of evolution of error minimization as a neutral consequence, or by-product, of the code expansion has been thoroughly developed in several recent studies [64,67,68]. Massey explored three distinct scenarios of the code expansion: the “2-1-3” model [78], which is similar to expansion schemes independently proposed by Higgs [79] and Francis [80], the ambiguity reduction model [81] and precursor-product scenario specified by the coevolution theory [57,59]. The 2-1-3 model and similar schemes posit that the code started from an ancestral stage, in which only the second base of the codon was informative, and expanded by assigning specificity to the first, and then, in some codon series, the third bases (another idea that goes back to Crick’s 1968 paper). Under the ambiguity reduction model, in the ancestral code, codon series ambiguously encoded groups of amino acids, such that the subsequent evolution involved gradual increase in the specificity of the codon-amino acid mapping. The evolutionary simulations with both the 2-1-3 model and the ambiguity reduction model readily yield codes with error minimization levels exceeding that of the SGC. In contrast, the coevolution model, although producing some level of error minimization, was substantially inferior to the other scenarios, as also demonstrated in an earlier analysis by Higgs [79]. These findings cast doubt on the direct relevance of coevolution model [57,59] and imply that, although substantial error minimization is a key property of the SGC, this feature likely evolved as a by-product of code expansion rather than by direct selection for code robustness.

## 5. Protein Evolution Paradox, Extinct Primordial Stereochemical Code, Expansion of the Code, and Frozen Accident: A Coevolution Scenario for the Code and the Translation System

The origin and evolution of the translation system is a forbiddingly difficult problem, and therefore, in many studies on the code evolution, it is formally treated as a separate issue and approached almost like a mathematical puzzle [9]. Ultimately, however, it appears virtually certain that evolution of the code can be understood only in the context of the evolution of translation, as presciently noted by Crick: “It is almost impossible to discuss the origin of the code without discussing the origin of the actual biochemical mechanisms of protein synthesis” [1].

The translation system is universally conserved among the extant cellular life forms but many protein components of the translation apparatus, as well as the tRNAs, are paralogs, and furthermore, belong to large paralogous families. Phylogenetic analysis of such families can provide clues to the pre-LUCA phase of evolution. The results of such analyses reveal a ‘protein evolution paradox’. The aminoacyl-tRNA synthetases (aaRS), the enzymes that are responsible for the accurate matching between amino acids and the cognate codons in the modern translation system, belong to two classes of paralogs, with 10 specificities in each [82,83]. As convincingly demonstrated for the Class I aaRS that contain a Rossmann-fold catalytic domain, the diversification of the aaRS occurred at a late stage in the evolution of the Rossmann-fold protein superfamily [84,85]. By the time the 10 Class I aaRS specificities evolved via radiation from a common ancestor, the evolution of the Rossmann fold superfamily had already produced a substantial diversity of other enzymatic and nucleotide-binding domains. A similar scenario holds for the evolution of the Class II aaRS, which belong to the biotin synthase superfamily, although the evolution of these proteins has not been explored as thoroughly as that of Class I [86,87]. For this early protein evolution to occur, high-fidelity (although possibly not to the level of the modern system) translation was certainly essential, and given that the different aaRS specificities have not yet evolved, the conclusion inevitably ensues that the specificity of amino acid-codon correspondence was determined by RNA molecules [88].

The conclusion that the mRNA decoding in the early translation system was performed by RNA molecules, conceivably, evolutionary precursors of modern tRNAs (proto-tRNAs) [89], implies a stereochemical model of code origin and evolution, but one that differs from the traditional models of this type in an important way (Figure 2). Under this model, the proto-RNA-amino acid interactions that defined the specificity of translation would not involve the anticodon (let alone codon) that therefore could be chosen arbitrarily and fixed through frozen accident. Instead, following the reasoning outlined previously [90], the amino acids would be recognized by unique pockets in the tertiary structure of the proto-tRNAs. The clustering of codons for related amino acids naturally follows from code expansion by duplication of the proto-tRNAs; the molecules resulting from such duplications obviously would be structurally similar and accordingly would bind similar amino acids, resulting in error minimization, in accord with Crick’s proposal (Figure 2). A variant of this scenario would involve initial imprecise recognition of groups of amino acids (e.g., several bulky hydrophobic ones) followed by duplication and subfunctionalization of the proto-tRNAs, recapitulating Woese’s statistical protein hypothesis [91]. This part of the scenario represents code expansion discussed above that could have been driven by the benefits of diversification of the repertoire of protein amino acids and would yield robustness as a byproduct.

The transition from the primordial, RNA only translation system to the modern RNA-protein machinery would follow the previously outlined path [90,92], in which the first translated peptides would serve as cofactors for the ribozymes, in particular, the proto-tRNAs (Figure 3). At the next stage, the ancestral catalytic domains of the aaRS would evolve and take over from the proto-tRNA as the catalysts for the aminoacylation reaction. Initially, each of these domains would indiscriminately catalyze aminoacylation of 10 proto-tRNAs. The subsequent evolution would involve duplication of the catalytic domains and accretion of accessory domains, resulting in the emergence of protein-mediated determination of aminoacylation specificity (Figure 3).

Once the amino acid specificity determinants shifted from the proto-tRNAs to the aaRS, the amino acid-binding pockets in the (proto) tRNAs deteriorated such that modern tRNAs showed no consistent affinity to the cognate amino acids. This scenario implies that attempts to decipher the primordial stereochemical code by comparative analysis of modern translation system components are largely futile. The best hope for reconstructing the ancient code could lie in experiments on in vitro evolution of specific aminoacylating ribozymes that can be evolved quite easily and themselves seem to recapitulate a key aspect of the primordial translation system [93,94,95]. It seems interesting to note that, although direct recognition of amino acids by RNA molecules seems to be banished from the modern translation system, it survives in other functional spheres of modern cells, in particular, as amino acid-binding riboswitches [96,97]. The structures of such riboswitches complexed with the cognate amino acids might shed light on the structures of primordial proto-tRNAs.

## 6. Concluding Remarks

Five decades after the publication of Crick’s seminal paper [1], frozen accident remains alive and well, and a major tenet of our thinking on the evolution of the code. It could be argued that frozen accident is not a theory and not even a hypothetical scenario but rather a meaningless non-explanation of the code origin. Although any detailed discussion on epistemology is beyond the scope of this article, I counter that random and (extremely) rare events are part and parcel of the evolution of life, even if this is unpalatable to some on philosophical grounds [88,98,99]. Thus, frozen accident actually is a falsifiable null hypothesis. The falsification of the frozen accident comes in the form of positive evidence of specific evolutionary processes that account for certain aspects of the code evolution. However, none of the three major theories of the code evolution has been fully successful in providing a definitive explanation although each has highlighted important features of the code. By combining findings from evolutionary analysis of translation system components, a distinct flavor of the stereochemical theory—the concept of the code expansion that is well grounded in data and theory—and the frozen accident idea, a simple, experimentally testable model of code evolution is proposed here. It owes most of the underlying ideas to Crick’s 1968 paper and the concurrent work of Carl Woese. 

## Figures and Tables

**Figure 1 life-07-00022-f001:**
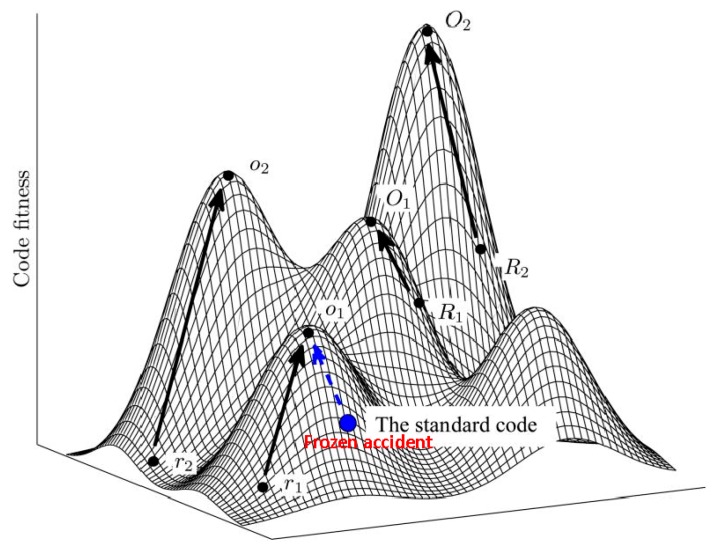
The code fitness landscape. The figure is a cartoon illustration of peaks of different heights separated by low fitness valleys on the code fitness landscape. The summit of each peak corresponds to a local optimum (O). Evolution towards local peaks is shown by arrows and starts either from a random code (R) or from the SGC. Modified from [14], under Creative License.

**Figure 2 life-07-00022-f002:**
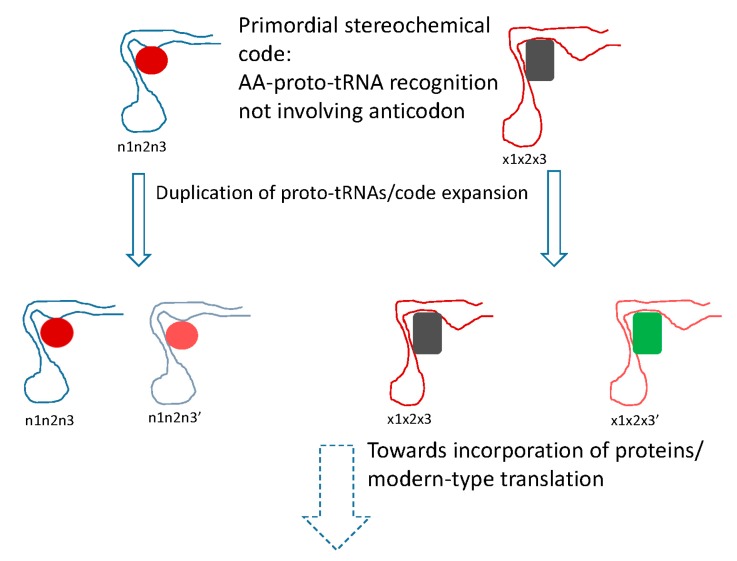
Putative primordial stereochemical code, evolution by code expansion and the standard code as frozen accident. Related amino acid (AA) are shown by similar colors. In the code expansion phase, the anticodons are shown changed in the third position (corresponding to the first position of the codon), resulting in recruitment of related amino acids.

**Figure 3 life-07-00022-f003:**
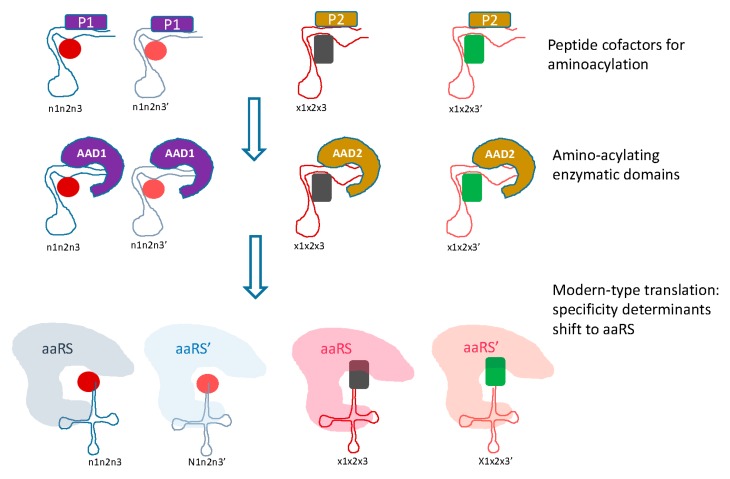
Transition from the RNA only to modern-type RNA-protein translation system. Two phases of evolution between the primordial, RNA only translation and the modern-type system are envisaged. At the first intermediate stage, peptides (denoted P) synthesized by the primitive translation system would serve as cofactors for amino acid binding and aminoacylation, and at the second stage, catalysis of aminoacylation would be relegated to catalytic aminoacylating domains (denoted AAD). Two forms of both the cofactor peptides and the AAD are shown, to provide continuity towards the two classes of aaRS.

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
