# Peer review of "Frozen Accident Pushing 50: Stereochemistry, Expansion, and Chance in the Evolution of the Genetic Code"

_life, 2017, doi:10.3390/life7020022_

Reviewer 1 Report

This paper reviewed the history of studies on the origin of genetic code since Crick’s paper celebrated by this special issue, and the alternative hypotheses of early evolution of genetic code. The author discussed various aspects on the problems of hypotheses of early evolution of genetic code. Then, finally, the author proposed his own hypothesis on the origin of genetic code. In this hypothesis, proto-tRNA was thought to have had specific affinity to certain amino acid, but its anticodon did not contribute amino acid recognition.

I was interested in this paper, and I think that many readers in this field should be interested in. However, I have several general comments on this paper.

1)     The author mentioned that the number of amino acids in early life was limited and less than 20. The author’s early amino acids are only based on the knowledge from meteorites. There are lots of studies on the prebiotic synthesis of amino acids since famous Miller’s experiment on 1953. For example, the author should refer and introduce the knowledge on the prebiotic synthesis of amino acids by Sutherland’s group.

2)     In meteorites and products of prebiotic synthesis experiments of amino acids, various amino acids that are not used in modern translation system are found. Therefore, one can imagine that the sort of amino acid in protein have decreased during evolution of early translation system. The author seemed to overlook this point. It is big mystery why only 20 (or 22) amino acids have been used in translation system, although there are a number of amino acids are known.

3)     As the example of study proposing stereochemical hypothesis of origin of genetic code, the author seems to refer only publications from Yarus’s group. However, other examples with certain experimental supports such as C4N hypothesis proposed by Shimizu (J. Mol. Evol. (1982) 18:, 297–303) that was supported by Watanabe & Miura (BBRC (1985) 129: 679-685) should be referred and be considered.

4)     On the coevolution theory. I think that the introduction of glutamine and asparagine is explained by this hypothesis. If it were thought to be unlikely, the author might be needed to show an alternative hypothesis.

5)     The author separated the anticodon sequence (codon recognition) and amino acid recognition, in his hypothetical “proto-tRNA” (Figure 2). I agree that the Yarus’ hypothesis (specific interactions between amino acids and codon/anticodon sequences observed in aptamer experiments) is not easy to accept, since both anticodons and codons are thought to interact with certain amino acid. The author’s hypothesis does not include this problem, but it is very difficult to find the experimental and/or theoretical supports on it. From this viewpoint, I request to the author to show or to suggest the experimental and/or theoretical supports on his hypothesis.

Minor suggestions are following.

P.3 line 56. “standard genetic“ should be “standard genetic code”.

P.4 line 80. “(Figure 1)”. I think that Figure 1 does not explain anything on this sentence.

P.9. lines 210-211. The exponents should be shown as superscripts. 

Author Response

1)          The author mentioned that the number of amino acids in early life was limited and less than 20. The author’s early amino acids are only based on the knowledge from meteorites. There are lots of studies on the prebiotic synthesis of amino acids since famous Miller’s experiment on 1953. For example, the author should refer and introduce the knowledge on the prebiotic synthesis of amino acids by Sutherland’s group.

Response: it is not the case that in the original manuscript the list of early amino acid was only based on meteorite data. Primordial chemistry experiments are mentioned explicitly, and the main source (admittedly, somewhat outdated) is Trifonov’s review of 2004 (Ref. 69 in the original). No attempt was intended here on a comprehensive review of such experiments. That said, the important work from the Sutherland lab is cited in the revision.

2)          In meteorites and products of prebiotic synthesis experiments of amino acids, various amino acids that are not used in modern translation system are found. Therefore, one can imagine that the sort of amino acid in protein have decreased during evolution of early translation system. The author seemed to overlook this point. It is big mystery why only 20 (or 22) amino acids have been used in translation system, although there are a number of amino acids are known.

Response: I am not sure there is a huge mystery here. The simplest amino acids on the early list are indeed the most abundant in the primordial chemistry experiments and, as far as I can see, in meteorites as well. I do not see a compelling reason to discuss this aspect in the present paper.

3)       As the example of study proposing stereochemical hypothesis of origin of genetic code, the author seems to refer only publications from Yarus’s group. However, other examples with certain experimental supports such as C4N hypothesis proposed by Shimizu (J. Mol. Evol. (1982) 18:, 297–303) that was supported by Watanabe & Miura (BBRC (1985) 129: 679-685) should be referred and be considered.

Response: There was much early effort on deciphering a ‘stereochemical code’, primarily through molecular modeling.  I cite some of this work, including Shimizu’s, in the revised manuscript. However, I am afraid that it is impossible to come up with defendable conclusions from these studies that are rather crude and poorly compatible with each other or the subsequent aptamer work. This was the conclusion of Knight et al. in their influential 1999 review, and I have to concur. All in all, the aptamer work remains the most credible series of studies relevant for the stereochemical hypothesis.

4)          On the coevolution theory. I think that the introduction of glutamine and asparagine is explained by this hypothesis. If it were thought to be unlikely, the author might be needed to show an alternative hypothesis.

Response: I am not sure that glutamine and asparagine are the best cases in point because for each, there are two radically different pathways of incorporation into protein. Regardless, I do not dismiss the coevolution scenario, far from it. It adds important constraints to any scheme of code evolution and is compatible with other scenarios. This should be particularly obvious from the revised manuscript in which such compatibility is specifically emphasized (see response to reviewer 2).

5)     The author separated the anticodon sequence (codon recognition) and amino acid recognition, in his hypothetical “proto-tRNA” (Figure 2). I agree that the Yarus’ hypothesis (specific interactions between amino acids and codon/anticodon sequences observed in aptamer experiments) is not easy to accept, since both anticodons and codons are thought to interact with certain amino acid. The author’s hypothesis does not include this problem, but it is very difficult to find the experimental and/or theoretical supports on it. From this viewpoint, I request to the author to show or to suggest the experimental and/or theoretical supports on his hypothesis.

 Response: the discussion of the hypothesis is expanded in the revision (see response to reviewer 2). The experiments required to test this hypothesis are outlined. Admittedly, the hypothesis is speculative whereas the experiments are not easy. However, I believe the hypothesis is logically justified as pointed out in the manuscript (particularly, the revised and expanded version). This is the best I can do in the context of a ‘concept article’.

Minor suggestions are following.

P.3 line 56. “standard genetic“ should be “standard genetic code”.

Response: corrected

P.4 line 80. “(Figure 1)”. I think that Figure 1 does not explain anything on this sentence.

Response: yes, the figure was mentioned here mistakenly; removed.

P.9. lines 210-211. The exponents should be shown as superscripts. 

Response: fixed

Reviewer 2 Report

This is a strange paper. The author suggests in his abstract that he intends to “combine a distinct version of the stereochemical hypothesis, in which amino acids are recognized via unique sites in the tertiary structure of proto-tRNAs, rather than by anticodons, expansion of the code via proto-tRNA duplication and the frozen accident.” That portion of the paper, which should be the “meat”, takes up only 15% of the  text (lines 269-317 of the paper) and is so thin as to be embarrassing. He provides no evidence whatsoever that proto-tRNAs have a diversity of conformations required to bind the ten or so primordial amino acids in a reasonably specific manner; provides no modeling of these structures or interactions (which is surely not beyond his ability to produce); and provides no rationale for how such specific tRNA-amino acid binding would yield a functional translation system. The final point is particularly troublesome since, in Figure 2, he literally jumps from a proto-tRNA world to a fully functioning aaRS world without a pause. I’m afraid I could only think of the famous cartoon where the scientist has one very complex equation on the left side of the blackboard and another, seemingly unrelated equation on the right, and has written in between, “And then a miracle happens!” to link the two. I presume the author has some ideas of how one gets from A to B without a miracle; it would be nice if, while he is speculating, he suggests his vision for how the rest of the gaps may have been filled in. That, combined with some modeling of proto-tRNA-aa binding would make an interesting paper.

I frankly think the rest of the paper can be shortened significantly or jettisoned. Yes, I know this is supposed to be volume celebrating Crick’s genetic code work, but frankly, Crick’s frozen accident theory isn’t technically a theory at all. It makes no testable predictions and places the issue of the code out of the reach of science. There is no point in bowing to nonsense of this sort. Hence the following criticisms:

Line 58: I find this line of reasoning completely unconvincing. This is a purely statistical argument that assumes that any sequence of nucleotides can theoretically encode any set of amino acids without restriction. The fact that there are ONLY billions of “better” codes should be a warning that the code is not an accident, but that the possibilities have been pared down VERY significantly from the 1084 possibilities.  The fact that the code is not absolutely optimized for error prevention (leaving billions of “better” possibilities) should similarly tell us that there are other considerations modifying selection.  And the fact that the number of variants from the existing code is limited almost exclusively to stop codon modifications should tell us that the combination of selection criteria is incredibly strenuous, preventing alternative codes from emerging.

Line 65: There can be no refutation of the frozen accident theory because it makes no testable predictions. This is basic logic. One cannot use the fact that we have only partially solved the origins of the genetic code so far to argue that accident is a substitute.

Line 127: This position completely undermines the whole point of the frozen accident argument. If, in fact, the code had already been NEARLY optimized prior to the “accidental freezing” of one of the possible variants, one must still explain how that one version drove the rest to extinction. 

Line 153: This is exactly that HGT does explain, without recourse to a “frozen accident” – HGT imposes a constraint that, in conjunction with others, yields a very small range of high-probability codes.

Line 166: Dividing the t heories into three competing approaches is exactly what’s wrong with origins-of-the-code research. There is NO reason to believe that any ONE of these explains the origins of the code and EVERY reason to believe that ALL THREE PLUS HGT are selection criteria that worked in tandem. This is a false trichotomy (tetrachotomy?).

Line 302: What does a frozen accident have to do with stereochemical specificity of proto-tRNAs for amino acids and the evolution of error minimization? Both obviate the need for an accident of any kind.

Figure 2: And then a miracle occurs! If one has a proto-tRNA that can bind specific  amino acids and participate in protein translation without the presence of aaRS, what is the evolutionary drive that gives rise to aaRS? You’ve promised us a way to get from an RNA world to a genetically encoded world, but you’ve skipped the critical step without even a mention of it!

In sum, I was very intrigued by the topic and I also find the idea of a proto-tRNA-aa intermediate translation system very plausible, but this paper has focused so much on trying to fit the idea into a frozen accident framework that it lost sight of what it really should do.

Author Response

This is a strange paper. The author suggests in his abstract that he intends to “combine a distinct version of the stereochemical hypothesis, in which amino acids are recognized via unique sites in the tertiary structure of proto-tRNAs, rather than by anticodons, expansion of the code via proto-tRNA duplication and the frozen accident.” That portion of the paper, which should be the “meat”, takes up only 15% of the  text (lines 269-317 of the paper) and is so thin as to be embarrassing. He provides no evidence whatsoever that proto-tRNAs have a diversity of conformations required to bind the ten or so primordial amino acids in a reasonably specific manner; provides no modeling of these structures or interactions (which is surely not beyond his ability to produce); and provides no rationale for how such specific tRNA-amino acid binding would yield a functional translation system. The final point is particularly troublesome since, in Figure 2, he literally jumps from a proto-tRNA world to a fully functioning aaRS world without a pause. I’m afraid I could only think of the famous cartoon where the scientist has one very complex equation on the left side of the blackboard and another, seemingly unrelated equation on the right, and has written in between, “And then a miracle happens!” to link the two. I presume the author has some ideas of how one gets from A to B without a miracle; it would be nice if, while he is speculating, he suggests his vision for how the rest of the gaps may have been filled in. That, combined with some modeling of proto-tRNA-aa binding would make an interesting paper.

Response: I do not believe that the percentage of the text dedicated to the presentation of the hypothesis is of any particular relevance. If the argument itself seems thin, that is a different matter. In the revised manuscript, I have expanded this part, in an attempt to outline ways in which some of the gaps can be filled. However, actual modeling of the binding is outside the scope of this ‘concept paper’.

I frankly think the rest of the paper can be shortened significantly or jettisoned. Yes, I know this is supposed to be volume celebrating Crick’s genetic code work, but frankly, Crick’s frozen accident theory isn’t technically a theory at all. It makes no testable predictions and places the issue of the code out of the reach of science. There is no point in bowing to nonsense of this sort.

Response: I am afraid that here, we come to irreconcilable differences. I do agree that frozen accident is technically not a theory (but neither is any of the code evolution scenarios). Crick’s repeated use of the term ‘theory’ is perhaps somewhat less than cautious. Clearly, though, he used ‘theory’ in the loose, vernacular sense, which is highly unadvisable (even if still common) in 2017, but I suspect was more tolerable in 1968. However, I strongly disagree with the claim that the frozen accident scenario “places the issue of the code out of the reach of science”. Not so, not unless any randomness is outside the purview of science, such that, for example, the role of genetic drift in evolution and the neutral theory (a genuine theory, in this case) should be banished, thus effectively destroying modern evolutionary biology. This claim stems from an unsupported and unsustainable epistemological stance. Certainly, anyone is entitled to their own epistemology but I decline to accept this as a legitimate part of a scientific debate. A brief comment on this was added in the Concluding Remarks Furthermore, I have to protest the characterization of frozen accident as ‘nonsense’, to which I supposedly ‘bow’. Had the ‘nonsense’ clause concerned my own reasoning only, I would have considered prudent to ignore it. However, I find it unconscionable that this language is applied to Crick’s seminal paper, and with no good reason at all. Thus, I decline to even consider shortening the manuscript,  let alone jettisoning the bulk of it on the basis of this argument. If it is considered legitimate, the entire paper has to be jettisoned.

Hence the following criticisms:

Line 58: I find this line of reasoning completely unconvincing. This is a purely statistical argument that assumes that any sequence of nucleotides can theoretically encode any set of amino acids without restriction. The fact that there are ONLY billions of “better” codes should be a warning that the code is not an accident, but that the possibilities have been pared down VERY significantly from the 1084 possibilities.  The fact that the code is not absolutely optimized for error prevention (leaving billions of “better” possibilities) should similarly tell us that there are other considerations modifying selection.  And the fact that the number of variants from the existing code is limited almost exclusively to stop codon modifications should tell us that the combination of selection criteria is incredibly strenuous, preventing alternative codes from emerging.

Response: Again, this is a pan-adaptationist argument: everything in biology is explained by combination  selection pressures. Everyone is entitled to their beliefs but no one is under obligation to take those seriously.

Line 65: There can be no refutation of the frozen accident theory because it makes no testable predictions. This is basic logic. One cannot use the fact that we have only partially solved the origins of the genetic code so far to argue that accident is a substitute.

Response: I am not sure I understand the invocation of basic logic. If frozen accident is accepted as null hypothesis, it can be partially refuted via positive evidence supporting selective or other mechanisms affecting its evolution. A brief comment to that effect included in the Concluding remarks.

Line 127: This position completely undermines the whole point of the frozen accident argument. If, in fact, the code had already been NEARLY optimized prior to the “accidental freezing” of one of the possible variants, one must still explain how that one version drove the rest to extinction.

Response:  I cannot see what is undermined here. Frozen accident claims that the codon assignments are random in and by themselves but do not change much or widely because that would be fatal. The structure of the codon table, optimization etc are different matters.

Line 153: This is exactly that HGT does explain, without recourse to a “frozen accident” – HGT imposes a constraint that, in conjunction with others, yields a very small range of high-probability codes.

Response: HGT  per se imposes no such constraint. What it does, is freezing a particular version of the code which could happen by drift, within the constraints of viability, of course.

Line 166: Dividing the t heories into three competing approaches is exactly what’s wrong with origins-of-the-code research. There is NO reason to believe that any ONE of these explains the origins of the code and EVERY reason to believe that ALL THREE PLUS HGT are selection criteria that worked in tandem. This is a false trichotomy (tetrachotomy?).

Response: with this point, I fully agree. Indeed, the tetrachotomy is false, and while I am not sure the original version of the manuscript made the impression that there could be one and only one scenario for the code evolution (at least, there was no such intent), stating this explicitly is useful. Such a statement was added in the revision: ‘To conclude this brief discussion of the three major scenarios for the evolution of the code, it is useful to note that, although stemming from widely different premises, there is no reason why these scenarios should be mutually exclusive. Quite the contrary, stereochemistry, biochemical coevolution, and selection for error minimization could have contributed synergistically at different stages of the evolution of the code {Knight, 1999 #1333} – along with frozen accident.’

Line 302: What does a frozen accident have to do with stereochemical specificity of proto-tRNAs for amino acids and the evolution of error minimization? Both obviate the need for an accident of any kind.

Response: not so. Within the proposed model at least, the codon assignments have everything to do with accident.

Figure 2: And then a miracle occurs! If one has a proto-tRNA that can bind specific  amino acids and participate in protein translation without the presence of aaRS, what is the evolutionary drive that gives rise to aaRS? You’ve promised us a way to get from an RNA world to a genetically encoded world, but you’ve skipped the critical step without even a mention of it!

Response: No miracles (or exclamation points) are needed, and I have not promised anything to anyone. Our understanding of the evolution of translation is poor but that said, a path from the RNA-based translation system to a modern-type one can proposed, and this is what I do in the revised manuscript and the new figure 3.

In sum, I was very intrigued by the topic and I also find the idea of a proto-tRNA-aa intermediate translation system very plausible, but this paper has focused so much on trying to fit the idea into a frozen accident framework that it lost sight of what it really should do.

Response: As pointed out above, there are irreconcilable differences here. I tried to address all the points in the review that I considered to be constructive and actionable.

Round  2

Reviewer 1 Report

This paper reviewed the history of studies on the origin of genetic code since Crick’s paper celebrated by this special issue. Then, finally, the author proposed his own hypothesis on the origin of genetic code. In this hypothesis, proto-tRNA was thought to have had specific affinity to certain amino acid, but its anticodon did not contribute amino acid recognition. In the revised manuscript, the author expanded his description of this hypothesis, and included some suggestion on the future experiment to study establishment of genetic code. I think that the author’s hypothesis become better working hypothesis for researchers using experimental approach for the evolution of genetic code.

I would like to note one question on the author’s hypothesis. The author implied the two AADs in his hypothesis as the origins of Class I and Class II ARS. Both Class I and Class II ARSs contains ARSs for positively charged amino acids, negatively charged amino acids, polar uncharged amino acids, and hydrophobic amino acids. Can we say the amino acids catalyzed by Class I ARSs have common characters which are different from those by Class II ARSs? 

Author Response

The reviewer asks an interesting question but, regrettably, the answer is: no. It does not seem possible to differentiate between amino acids that are activated by Class I and Class II aaRS by any simple physico-chemical criteria. Therefore, I do not find it necessary to discuss this issue in the present manuscript (I do not believe the reviewer actually strongly suggests it is).

Reviewer 2 Report

While the author and I have irreconcileable philosophical differences, the revisions substantially address these and provide a much clearer presentation of the author's proposal.While I am still not convinced, I very much appreciate the author's willingness to address the issues and clarify critical points. This clarity will help define future developments in the field.

Author Response

I truly appreciate the constructive and collegiate position of the reviewer.